# High Sedentary Behavior Is Associated with Depression among Rural South Africans

**DOI:** 10.3390/ijerph16081413

**Published:** 2019-04-19

**Authors:** Supa Pengpid, Karl Peltzer

**Affiliations:** 1ASEAN Institute for Health Development, Mahidol University, Salaya, Phutthamonthon, Nakhonpathom 73170, Thailand; supaprom@yahoo.com; 2Deputy Vice Chancellor Research and Innovation Office, North West University, Potchefstroom 2531, South Africa

**Keywords:** sedentary behavior, depression, confounding factors, adults, rural South Africa, HAALSI

## Abstract

The study aimed to investigate the association between sedentary behavior and depression among rural South Africans. Data were analyzed from the cross-sectional baseline survey of the “Health and Aging in Africa: A Longitudinal Study of an INDEPTH community in South Africa (HAALSI)”. Participants responded to various measures, including sociodemographic information, health status, anthropometric measures, and sedentary behavior. The sample included 4782 persons (40 years and above). Overall, participants engaged in <4 h (55.9%), 4–<8 h (34.1%), 8–<11 h (6.4%), or 11 or more h a day (3.5%) of sedentary behavior, and 17.0% screened positive for depression. In multivariable logistic regression, which was adjusted for sociodemographic variables (Model 1) (Odds Ratio, or OR: 2.45, Confidence Interval, or CI: 1.74, 3.46) and adjusted for sociodemographic and health variables, including physical activity (Model 2) (OR: 3.00, CI: 2.00, 4.51), high sedentary time (≥11 h) was independently associated with depression. In combined analysis, compared to persons with low or moderate sedentary behavior (<8 h) and moderate or high physical activity, persons with high sedentary behavior (≥8 h) and low physical activity were more likely to have depression in Model 1 (OR: 1.60, CI: 1.65, 3.13) and Model 2 (OR: 1.60, CI: 1.05, 2.44). Findings support that sedentary behavior and combined sedentary behavior and low physical activity may be a modifiable target factor for strategies to reduce depression symptoms in this rural population in South Africa.

## 1. Introduction

Sedentary behavior (low energy expenditure in reclining, sitting, or lying position) has been conceptualized as being different from physical inactivity, and constitutes an “independent predictor of metabolic risk even if an individual meets current physical activity guidelines” [1]. Based on a systematic review, sedentary behavior has been linked to mortality and non-communicable disease morbidity [2]. Likewise, depression contributes to a large extent to the “global burden of disease” [3]. A systematic review, mainly in high-income countries, found that sedentary behavior increases the risk for depression [4]. In a recent study in Japan, no significant associations were found between sedentary behavior levels and depression [5]. 

Interventions to reduce sedentary behavior may reduce depression [6]. There are limited studies investigating sedentary behavior and depression in low-income and middle-income countries. Exceptions include a study among adults in Brazil that found more than 5 h of television (TV) viewing per day was associated with a higher risk for depression [7], and among predominantly older adults in five middle-income countries that found that persons with depression have a higher risk for higher levels of sedentary behavior [8]. The aim of this investigation was to assess the association between sedentary behavior and depression among rural South Africans.

## 2. Methods

### 2.1. Sample and Procedure

The study included 4782 individuals with complete sedentary behavior measurements, which had a response rate 85.9%, from “Health and Aging in Africa: A Longitudinal Study of an INDEPTH Community in South Africa (HAALSI)” in the INDEPTH Health and Demographic Surveillance System (HDSS) site of Agincourt in 2015 in rural South Africa [9]. Sampling and procedures have been previously described [9,10]. The study protocol was approved the “University of the Witwatersrand Human Research Ethics Committee (ref. M141159), the Harvard T. H. Chan School of Public Health, Office of Human Research Administration (ref. C13–1608–02), and the Mpumalanga Provincial Research and Ethics Committee” [9]. Prior to assessments, participants signed an informed consent. 

### 2.2. Measures

#### 2.2.1. Exposure Variables

Sedentary behavior was assessed with two items asking about sedentary time in h and min. The question was asked for both a typical weekday and weekend day [9,11]. Sedentary time was classified into <4 h, 4–<8 h, 8–<11 h, and ≥11 h a day [12]. 

#### 2.2.2. Outcome Variable

Depression was assessed with the Center for Epidemiological Studies—Depression Scale (CES-D) eight-item questionnaire, with “a cut-off of three or more symptoms signifying depressive symptoms” [13] (Cronbach’s alpha 0.66). A previous study found the CES-D-10 to be a “valid and reliable screening scale for depression in South Africa” [14].

#### 2.2.3. Confounding Variables

Sociodemographic information included sex, age, a wealth index, and formal education [9,10]. 

Social cohesion was measured with four questions (e.g., “Most people in this village are willing to help their neighbors”). Response options ranged from 1 = strongly agree to 4 = strongly disagree [9] (Cronbach’s alpha 0.87). Social cohesion was reverse scored such that higher scores indicated more social cohesion (7–11 = low, 12 = medium, 13–16 = high) [10]. 

Current tobacco use was measured with two questions on smoking and smokeless tobacco use [9,10]. 

Alcohol dependence was assessed with the Cut down, Annoyed, Guilty, Eye opener (CAGE) questionnaire [15] (Cronbach’s alpha 0.82).

Physical activity was assessed with the General Physical Activity Questionnaire (GPAQ) [16,17]. Scores were grouped into low, moderate, and high physical activity according to the following GPAQ guidelines [17]. “High physical activity: a person reaching any of the following criteria is classified in this category: vigorous-intensity activity on at least three days achieving a minimum of at least 1500 MET (metabolic equivalent)-min per week OR; seven or more days of any combination of walking, moderate, or vigorous intensity activities achieving a minimum of at least 3000 MET-min per week. Moderate physical activity: a person not meeting the criteria for the ‘high’ category, but meeting any of the following criteria is classified in this category: three or more days of vigorous-intensity activity of at least 20 min per day OR; five or more days of moderate-intensity activity or walking of at least 30 min per day OR; five or more days of any combination of walking, moderate, or vigorous intensity activities achieving a minimum of at least 600 MET-min per week. Low physical activity: a person not meeting any of the above-mentioned criteria falls in this category.” [17].

Body Mass Index (BMI) was assessed from objective measures of height and body weight [9], and classified based on World Health Organization (WHO) criteria into underweight (<18.5 kg/m^2^), normal weight (18.5–24.9 kg/m^2^), overweight (25–29.9 kg/m^2^), and obese (30+ kg/m^2^) [10,18].

Chronic conditions included self-reported Human Immunodeficiency Virus (HIV) status, angina, heart failure, stroke, kidney disease, myocardial infarction, cataract, tuberculosis, measured hypertension, dyslipidemia, diabetes, anemia [19], and symptom-based chronic bronchitis [9,20]. All 13 chronic conditions were summed and divided into zero, one, two, or more.

Activities of daily living (ADLs) “measures included difficulty in walking, eating, bathing, getting in/out of bed, and using the toilet” [9]. Functional disability was defined as one or more positive responses to the six ADLs.

## 3. Data Analysis

Logistic regression was used to estimate the associations between sedentary behavior and depression. Model 1 was adjusted for sociodemographic variables, and Model 2 was adjusted for sociodemographic and health variables, including physical activity levels. Moreover, we used multivariable logistic regression to estimate the combined relationship between sedentary behavior and physical activity with depression. For the combined logistic regression analysis, the sample was subdivided based on sedentary and physical activity levels into four groups: (1) low or moderate sedentary time (<8 h) plus moderate or high physical activity group (reference category), (2) low or moderate sedentary time (<8 h) plus low physical activity group, (3) high sedentary time (≥8 h) plus moderate or physical activity group, and (4) high sedentary time (≥8 h) plus low physical activity. *p* < 0.05 was considered significant. All the statistical procedures were conducted with STATA software version 14 (Stata Corporation, College Station, TX, USA).

## 4. Results

### 4.1. Sample Characteristics

The sample consisted of 4782 persons that were 40 years and older (median 61 years, interquartile range = 20 years) and largely African Shangaan/Tsonga-speaking. More than half (53.3%) were female, 44.7% had no formal education, 93.2% had moderate to high social cohesion, and 58.0% were overweight or obese. More than half of participants (55.7%) had two or more chronic conditions, 8.7% had a functional disability, 15.6% were current tobacco users, and 1.4% had alcohol dependence. Overall, participants engaged in <4 h (55.9%), 4–<8 h (34.1%), 8–<11 h (6.4%), and ≥11 h per day (3.5%) of sedentary time, while 42.1% engaged in low physical activity, and 17.0% screened positive for depression (see Table 1).

### 4.2. Odds Ratios for Depression According to Levels of Sedentary Behavior

In multivariable logistic regression adjusted for sociodemographic variables (Model 1) (Odds Ratio, or OR: 2.45, Confidence Interval, or CI: 1.74, 346) and adjusted for sociodemographic and health variables, including physical activity (Model 2) (OR: 3.00, CI: 2.00, 4.51), high sedentary time (≥11 h) was independently associated with depression (see Table 2).

### 4.3. Odds Ratios for Depression According to Combined Sedentary Behavior and Low Physical Activity Behavior

In adjusted logistic regression analysis, compared to persons with low or moderate sedentary behavior (<8 h) and moderate or high physical activity, persons with high sedentary behavior (≥8 h) and low physical activity were more likely to have depression in Model 1 (OR: 1.60, CI: 1.65, 3.13) and Model 2 (OR: 1.60, CI: 1.05, 2.44). In addition, compared to persons with low or moderate sedentary behavior (<8 h) and moderate and high physical activity, persons with high sedentary behavior (≥8 h) and moderate or high physical activity were more likely to have depression in Model 1 (OR: 3.17, CI: 2.01, 5.00) and Model 2 (OR: 3.06, CI: 1.82, 5.12) (see Table 3).

## 5. Discussion

The study aimed to investigate associations between sedentary behavior and depression among rural South Africans. Consistent with a previous review [3] and other recent studies in middle-income countries [7,8], the study found that higher sedentary time (≥11 h) increased the odds for depression, after adjustments with relevant confounding factors. In the previous study of six middle-income countries, the higher risk for depression was found with a high sedentary time of 8 or more h, and particularly for 11 or more h [8]. The significantly increased risk for depression with a cut-off of 8 h in the previous study [8] may be due to the much larger sample size (*N* = 42,469) compared to this study (*N* = 4782). In Model 1 of our adjusted analysis with sociodemographic confounders, a high sedentary time of 8 to <11 h was marginally significantly associated with depression. In a study in Japan in the general adult population [5], 6 or more h of sedentary behavior was not associated with depression in adjusted analysis. Analyzing our data with sedentary time as a continuous variable, the significant relationship between increasing sedentary time and depression persisted. Moreover, the study found a combined association of high sedentary time (≥8 h) and low physical activity with depression. This finding is conforming to a study among Japanese adults [5]. Mechanisms on the relationship between sedentary behavior and depression are not so clear [21]. Several possible mechanisms have been proposed, e.g., that “autonomic and inflammatory responses to stress may be heightened in sedentary individuals contributing to risk in mood disturbances independent of reduction in physical activity” [21]. Possible health policy and practice implications include that the reduction of sedentary behavior and the reduction of combined sedentary behavior and low physical activity could reduce depression levels. 

The study found differences in the association between sedentary behavior and depression among different age groups, such that the depression risk from sedentary behavior was higher among older participants (60 years and older) than among those of middle-age (40–59 years). This could mean that efforts to prevent depression as a consequence of sedentary behavior could focus on older age groups. In addition, having a functional disability and/or having two or more chronic conditions were associated with depression. In a previous study [8], functional disability mediated 49% of the association between sedentary behavior and depression. All the more, it would be important to introduce interventions that reduce or interrupt sedentary time targeting individuals with functional disability and chronic conditions. 

## 6. Study Limitations

The study was limited to cross-sectional data in one sub-district in Mpumalanga province, South Africa. Therefore, no causative conclusions can be made, and the results cannot be generalized beyond the studied sub-district. Sedentary behavior was measured by self-report, and future studies should include both self-report and objective measures. The assessment of the outcome variable, depression, used only a screening measure, and future studies may include diagnostic interviews.

## 7. Conclusions

Findings support that sedentary behavior and combined sedentary behavior and low physical activity may be a modifiable target factor for strategies to reduce depression symptoms in this rural population in South Africa.

## Figures and Tables

**Table 1 ijerph-16-01413-t001:** Sample characteristics.

Variable	Sample	Sedentary time/day				Depression
		<4 h	4–<8 h	8–<11 h	≥11 h	
	*N* (%)	*N* (%)	*N* (%)	*N* (%)	*N* (%)	*N* (%)
Sociodemographics						
All Age (years) 40–49 50–59 60–69 70 or more	4782 885 (18.5) 1349 (28.2) 1226 (25.6) 1322 (27.6)	2675 (55.9) 519 (58.6) 802 (59.5) 666 (54.3) 446 (53.9)	1632 (34.1) 309 (34.9) 433 (32.1) 444 (36.2) 276 (33.3)	306 (6.4) 40 (4.5) 82 (6.1) 73 (6.0) 67 (8.1)	169 (3.5) 17 (1.9) 32 (2.4) 43 (3.5) 39 (4.7)	795 (17.0) 91 (10.5) 180 (13.5) 220 (18.3) 304 (24.2)
Sex Female Male	2549 (53.3) 2233 (46.4)	1466 (57.5) 1209 (54.1)	830 (32.8) 795 (35.6)	151 (5.9) 155 (6.9)	95 (3.7) 74 (3.3)	460 (18.5) 335 (15.4)
Formal education None Grade 1–7 Grade 8–11 Grade 12 or more	2130 (44.7) 1653 (34.7) 555 (11.6) 430 (9.0)	1250 (58.7) 865 (52.3) 306 (55.1) 243 (56.5)	658 (30.9) 613 (37.1) 206 (37.1) 152 (35.3)	141 (6.6) 110 (6.7) 30 (5.4) 25 (5.8)	81 (3.8) 65 (3.9) 13 (2.3) 10 (2.3)	393 (19.2) 296 (18.1) 69 (12.5) 37 (8.7)
Wealth index Poor Second poorest Medium Second richest Rich	984 (20.6) 945 (19.8) 931 (19.5) 949 (19.8) 973 (20.3)	599 (60.9) 533 (56.4) 492 (52.8) 528 (55.6) 523 (53.8)	305 (31.0) 303 (32.1) 329 (35.3) 328 (34.6) 367 (37.7)	58 (5.9) 65 (6.9) 68 (7.3) 56 (5.9) 59 (6.1)	22 (2.2) 44 (4.7) 42 (4.5) 37 (3.9) 24 (2.5)	160 (16.8) 177 (19.2) 177 (19.5) 146 (15.7) 135 (14.1)
Social cohesion Low Moderate High	317 (6.8) 2383 (51.1) 1967 (42.1)	193 (60.9) 1219 (51.2) 1211 (61.6)	104 (32.8) 846 (35.5) 642 (32.6)	9 (2.8) 209 (8.8) 70 (3.6)	11 (3.5) 109 (4.6) 44 (2.2)	92 (29.1) 443 (18.6) 259 (13.2)
Health variables						
Body Mass Index (BMI) (kg/m^2^) 18.5 to <25 <18.5 25 to <30 ≥30	1617 (36.5) 243 (5.5) 1261 (28.5) 1311 (29.5)	909 (56.2) 125 (51.4) 740 (58.7) 735 (56.1)	557 (34.4) 94 (38.7) 391 (31.0) 469 (35.8)	95 (5.9) 16 (6.6) 82 (6.5) 67 (5.1)	56 (3.5) 8 (3.3) 48 (3.8) 40 (3.1)	230 (14.5) 52 (22.3) 212 (17.0) 211 (16.2)
Number of chronic conditions 0 1 ≥2	472 (12.3) 1225 (32.0) 2137 (55.7)	254 (53.8) 668 (54.5) 1209 (56.6)	175 (37.1) 448 (36.6) 709 (33.2)	32 (6.8) 70 (5.7) 134 (6.3)	11 (2.3) 39 (3.2) 85 (4.0)	48 (10.4) 157 (13.0) 423 (20.2)
Functional disability	413 (8.7)	194 (47.0)	121 (29.3)	68 (16.5)	30 (7.3)	139 (38.6)
Current tobacco use	743 (15.6)	392 (52.8)	276 (37.1)	46 (6.2)	29 (3.9)	146 (20.2)
Alcohol dependence	65 (1.4)	33 (50.8)	28 (43.1)	3 (4.6)	1 (1.5)	10 (15.9)
Physical activity High Moderate Low	1630 (34.3) 1122 (23.6) 2001 (42.1)	1051 (64.5) 524 (46.7) 1081 (54.0)	485 (29.8) 458 (40.8) 683 (34.1)	53 (3.3) 109 (9.7) 142 (7.1)	41 (2.5) 31 (2.8) 95 (4.7)	235 (14.6) 145 (13.1) 405 (21.2)
Sedentary time a day <4 h 4–<8 h 8–<11 h ≥11 h	2675 (55.9) 1632 (34.1) 306 (6.4) 169 (3.5)					427 (16.3) 242 (15.2) 65 (22.6) 61 (37.0)
Combined sedentary behavior and physical activity						
Sedentary low or moderate (<8 h) and physical activity (PA) moderate or high	1536 (32.3)					204 (13.4)
Sedentary low/moderate (<8 h) and PA low	1764 (37.1)					338 (20.0)
Sedentary high (≥8 h) and PA moderate or high	94 (2.0)					31 (33.3)
Sedentary high (≥8 h) and PA low	237 (5.0)					67 (30.7)

**Table 2 ijerph-16-01413-t002:** Odds ratios for depression according to levels of sedentary behavior.

Variable	Model 1 AOR (95% CI)	*p*-Value	Model 2 AOR (95% CI)	*p*-Value
Sedentary time a day				
<4 h	1 (Reference)		1 (Reference)	
4–<8 h	0.89 (0.75, 1.07)	0.210	0.94 (0.76, 1.16)	0.563
8–<11 h	1.35 (1.00, 1.83)	0.052	1.19 (0.80, 1.77)	0.388
≥11 h	2.45 (1.74, 3.46)	<0.001	3.00 (2.00, 4.51)	<0.001
Sociodemographics				
Age (years) 40–49 50–59 60–69 70 or more	1 (Reference) 1.25 (0.95, 1.65) 1.76 (1.32, 2.34) 2.41 (1.82, 4.20)	0.119 <0.001 <0.001	1 (Reference) 1.01 (0.73, 1.41) 1.66 (1.19, 2.30) 1.92 (1.37, 2.70)	0.942 0.003 <0.001
Sex Female Male	1 (Reference) 0.79 (0.68, 0.93)	0.005	1 (Reference) 0.79 (0.64, 0.95)	0.015
Formal education None Grade 1–7 Grade 8–11 Grade 12 or more	1 (Reference) 1.15 (0.96, 1.38) 0.92 (0.68, 1.25) 0.72 (0.48, 1.08)	0.141 0.581 0.115	1 (Reference) 1.23 (0.99, 1.53) 1.02 (0.72, 1.46) 0.78 (0.48, 1.28)	0.068 0.916 0.323
Wealth index Poor Second poorest Medium Second richest Rich	1 (Reference) 1.12 (0.88, 1.43) 1.20 (0.94, 1.53) 0.92 (0.71, 1.19) 0.92 (0.70, 1.21)	0.349 0.150 0.512 0.541	1 (Reference) 0.92 (0.69, 1.24) 1.05 (0.78, 1.41) 0.84 (0.62, 1.14) 0.90 (0.66, 1.24)	0.513 0.760 0.344 0.580
Social cohesion Low Moderate High	1 (Reference) 0.53 (0.40, 0.69) 0.36 (0.28, 0.48)	<0.001 <0.001	1 (Reference) 0.64 (0.45, 0.90) 0.43 (0.30, 0.61)	0.013 <0.001
Health variables				
Body Mass Index (BMI) (kg/m^2^) 18.5 to <25 <18.5 25 to <30 ≥30			1 (Reference) 1.62 (1.08, 2.45) 1.19 (0.94, 1.51) 1.17 (0.91, 1.50)	0.016 0.190 0.304
Number of chronic conditions 0 1 ≥2			1 (Reference) 1.29 (0.89, 1.86) 2.05 (1.45, 2.89)	0.196 <0.001
Functional disability			2.02 (1.45, 2.80)	<0.001
Current tobacco use			1.16 (0.88, 1.51)	0.294
Alcohol dependence			1.25 (0.54, 2.90)	0.596
Physical activity High Moderate Low			1 (Reference) 0.89 (0.69, 1.14) 1.01 (0.81, 1.26)	0.350 0.943

AOR = Adjusted Odds Ratio; CI = Confidence Interval; Model 1: Adjusted for sociodemographic variables; Model 2: Adjusted for sociodemographic and health variables.

**Table 3 ijerph-16-01413-t003:** Odds ratios for depression according to combined sedentary behavior and low physical activity.

Variable	Model 1 AOR (95% CI)	*p*-Value	Model 2 AOR (95% CI)	*p*-Value
Sedentary low or moderate and physical activity (PA) moderate or high	1 (Reference)		1 (Reference)	
Sedentary low/moderate and PA low	1.44 (1.22, 1.71)	<0.001	1.08 (0.88, 1.33)	0.442
Sedentary high and PA moderate or high	3.17 (2.01, 5.00)	<0.001	3.06 (1.82, 5.12)	>0.001
Sedentary high and PA low	2.27 (1.65, 3.13)	<0.001	1.60 (1.05, 2.44)	0.029
Sociodemographics				
Age (years) 40–49 50–59 60–69 70 or more	1 (Reference) 1.24 (0.94, 1.63) 1.68 (1.26, 2.24) 2.21 (1.66, 2.95)	0.139 <0.001 <0.001	1 (Reference) 1.00 (0.72, 1.40) 1.59 (1.14, 2.21) 1.87 (1.33, 2.63)	0.981 0.006 <0.001
Sex Female Male	1 (Reference) 0.80 (0.68, 0.94)	0.006	1 (Reference) 0.77 (0.63, 0.94)	0.011
Formal education None Grade 1–7 Grade 8–11 Grade 12 or more	1 (Reference) 1.15 (0.96, 1.38) 0.94 (0.69, 1.27) 0.74 (0.50, 1.11)	0.139 0.675 0.151	1 (Reference) 1.21 (0.98, 1.51) 1.03 (0.72, 1.47) 0.79 (0.48, 1.29)	0.081 0.891 0.338
Wealth index Poor Second poorest Medium Second richest Rich	1 (Reference) 1.12 (0.88, 1.44) 1.21 (0.95, 1.55) 0.92 (0.69, 1.18) 0.90 (0.69, 1.18)	0.354 0.129 0.520 0.451	1 (Reference) 0.93 (0.69, 1.24) 1.06 (0.79, 1.42) 0.86 (0.63, 1.17) 0.91 (0.66, 1.24)	0.606 0.704 0.341 0.534
Social cohesion Low Moderate High	1 (Reference) 0.52 (0.40, 0.69) 0.37 (0.28, 0.49)	<0.001 <0.001	1 (Reference) 0.64 (0.45, 0.90) 0.44 (0.31, 0.62)	0.011 <0.001
Health variables				
Body Mass Index (BMI) (kg/m^2^) 18.5 to <25 <18.5 25 to <30 ≥30			1 (Reference) 1.67 (1.11, 2.52) 1.19 (0.94, 1.52) 1.15 (0.89, 1.47)	0.015 0.151 0.293
Number of chronic conditions 0 1 ≥2			1 (Reference) 1.30 (0.90, 1.88) 2.04 (1.45, 2.89)	0.163 <0.001
Functional disability			2.11 (1.52, 2.94)	<0.001
Current tobacco use			1.19 (0.91, 1.56)	0.216
Alcohol dependence			1.27 (0.55, 2.94)	0.570

AOR = Adjusted Odds Ratio; CI = Confidence Interval; Model 1: Adjusted for sociodemographic variables; Model 2: Adjusted for sociodemographic and health variables.

## Data Availability

The baseline data used in this study is publicly available at the Harvard Center for Population and Development Studies (HCPDS) program website (www.haalsi.org).

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
