# Peer review of "High Sedentary Behavior Is Associated with Depression among Rural South Africans"

_ijerph, 2019, doi:10.3390/ijerph16081413_

Reviewer 1 Report
Comments:
This is good to know that a longitudinal survey on health and aging was initiated in Africa and the baseline data seems to include a lot of relevant information. This paper assessed the relationship between sedentary behaviour and depression among the older cohort and concluded with a very interesting finding. However, the analytic approach should be improved prior to publication:
Major revision:
1) Please specify clearly whether the data is only collected from rural areas of South Africa or the authors intended to include respondents in rural areas of South Africa only. If the latter, explain the reasons.
2) Please add more advanced statistical models such as path analysis to investigate the linkage of the topic concerned. The authors could utilize the dataset and provide strong scientific evidence on the findings using more sophisticated models.
3) This is good to explore the cut-off of the sedentary time by using ROC curve.
4) please state clearly the definition of low, moderate and high physical activity variables. Also, the definition of functional disability.
5) the study concluded that the sedentary time of 11 or more hours increased risk for depression and suggest for intervention to reduce sendentary time specifically for older people with functional disability and chronic condition. It may be good to add the occupational types or employment status to depict the situation for age 40-69 and suggest ways for the type of interventions.
Minor revision:
1) please delete the unnecessary double quotes.
2) please paraphrase and use appropriate word to connect the sentences in the first and second paragraph.
Author Response
This is good to know that a longitudinal survey on health and aging was initiated in Africa and the baseline data seems to include a lot of relevant information. This paper assessed the relationship between sedentary behaviour and depression among the older cohort and concluded with a very interesting finding. However, the analytic approach should be improved prior to publication:
Major revision:
1)Please specify clearly whether the data is only collected from rural areas of South Africa or the authors intended to include respondents in rural areas of South Africa only. If the latter, explain the reasons.
Response: Data were only collected from one rural site in South Africa
2) Please add more advanced statistical models such as path analysis to investigate the linkage of the topic concerned. The authors could utilize the dataset and provide strong scientific evidence on the findings using more sophisticated models.
Response: we wanted to use analysis methods, which have been commonly used, but we have added a combined analysis of sedentary behaviour and low physical activity
3) This is good to explore the cut-off of the sedentary time by using ROC curve.
Response: we did not get a good result
4) please state clearly the definition of low, moderate and high physical activity variables. Also, the definition of functional disability.
Response: both are added
5) the study concluded that the sedentary time of 11 or more hours increased risk for depression and suggest for intervention to reduce sendentary time specifically for older people with functional disability and chronic condition. It may be good to add the occupational types or employment status to depict the situation for age 40-69 and suggest ways for the type of interventions.
Response: do not have occupational types, only employment status (74.2% unemployed)
Minor revision:
1)please delete the unnecessary double quotes.
Response: Corrected
2) please paraphrase and use appropriate word to connect the sentences in the first and second paragraph.
Response: Corrected
Reviewer 2 Report
Comments:
This is an interesting study. For completeness I am surprised there is no information about ethnicity
There are really very few people reporting sedentary activity beyond 8 hours per day and they have characteristics of poor general health. It would be interesting and useful to have pen pictures of who they are and what they are like - fleshing out the statistics, if that were possible. The question of chicken and egg might be raised to discuss
There are intervention studies which demonstrate that increasing activity/reducing sedentary behaviour improves mood. Whether this is possible in the face of pathologies is another question
Author Response:
This is an interesting study. For completeness I am surprised there is no information about ethnicity.
Response: added as below
largely African Shangaan/Tsonga-speaking
There are really very few people reporting sedentary activity beyond 8 hours per day and they have characteristics of poor general health. It would be interesting and useful to have pen pictures of who they are and what they are like - fleshing out the statistics, if that were possible. The question of chicken and egg might be raised to discuss
Response: In Table 1 the sample characteristics are now stratified by sedentary time categories
There are intervention studies which demonstrate that increasing activity/reducing sedentary behaviour improves mood. Whether this is possible in the face of pathologies is another question
Response: this is referred to in the introduction {reference 6}
Round 2
Reviewer 1 Report
Comments:
Minor revisions are needed:
1) In paragraph one: “In a recent study in Japan, no significant associations were found between sedentary behaviour levels and depression [5]. Interventions to reduce sedentary behaviour may reduce depression [6].”
No significant but intervention to reduce sendentary behaviour? Should the intervention sentence place in the next paragraph?
2) please correct the reporting error in ODD ratio of the combined model 1.
3) In the combined models 1 and 2, please recheck the odd ratio and p-value of the low sendentary behaviour but high/moderate physical activity. Implication of significant findings should also be highlighted.
Author Response
Minor revisions are needed:
1) In paragraph one: “In a recent study in Japan, no significant associations were found between sedentary behaviour levels and depression [5]. Interventions to reduce sedentary behaviour may reduce depression [6].”
No significant but intervention to reduce sendentary behaviour? Should the intervention sentence place in the next paragraph?
Response: moved accordingly
2) please correct the reporting error in ODD ratio of the combined model 1.
Response: Corrected
3) In the combined models 1 and 2, please recheck the odd ratio and p-value of the low sendentary behaviour but high/moderate physical activity.
Response: the low/moderate sedentary and moderate/high physical activity is the reference category. The other odds ratios and p-values were rechecked.
Implication of significant findings should also be highlighted.
Response: below is added. Possible health policy and practice implications include that the reduction of sedentary behaviour and the reduction of combined sedentary behaviour and low physical activity could reduce depression levels.